# Protective Effect of Cashew Gum (*Anacardium occidentale* L.) on 5-Fluorouracil-Induced Intestinal Mucositis

**DOI:** 10.3390/ph12020051

**Published:** 2019-04-03

**Authors:** João Antônio Leal de Miranda, João Erivan Façanha Barreto, Dainesy Santos Martins, Paulo Vitor de Souza Pimentel, Deiziane Viana da Silva Costa, Reyca Rodrigues e Silva, Luan Kelves Miranda de Souza, Camila Nayane de Carvalho Lima, Jefferson Almeida Rocha, Ana Paula Fragoso de Freitas, Durcilene Alves da Silva, Ariel Gustavo Scafuri, Renata Ferreira de Carvalho Leitão, Gerly Anne de Castro Brito, Jand Venes Rolim Medeiros, Gilberto Santos Cerqueira

**Affiliations:** 1Department of Morphology, Faculty of Medicine, Federal University of Ceará, s/n Delmiro de Farias Street, Porangabuçu Campus, Fortaleza 60416-030, Brazil; erivanfacanha@yahoo.com.br (J.E.F.B.); dainy.santos@gmail.com (D.S.M.); paulo_vitordesouza@hotmail.com (P.V.d.S.P.); deiziane2009@gmail.com (D.V.d.S.C.); paulinhaff2@hotmail.com (A.P.F.d.F.); urologia@gmail.com (A.G.S.); leitao_renata@yahoo.com.br (R.F.d.C.L.); gerlybrito@hotmail.com (G.A.d.C.B.); giufarmacia@hotmail.com (G.S.C.); 2Biotechnology and Biodiversity Center Research, BIOTEC, Federal University of Piauí, Parnaíba, Piauí 64202-020, Brazil; reyca_14@hotmail.com (R.R.e.S.); luankelves11@gmail.com (L.K.M.d.S.); durcileneas@yahoo.com.br (D.A.d.S.); jandvenes@ufpi.edu.br (J.V.R.M.); 3Nucleus of Research and Development of Medications (NPDM), Federal University of Ceará, Coronel Nunes de Melo Street, 100, Fortaleza 60430-275, Brazil; camilacarvalhoenf@yahoo.com.br; 4Research Group in Natural Sciences and Biotechnology, Federal University of Maranhão, s/n Avenue Aurila Maria Santos Barros de Sousa, Frei Alberto Beretta, Grajaú-MA 65940-000, Brazil; jeffersonkalel@hotmail.com

**Keywords:** intestinal mucositis, heteropolysaccharide, 5-fluorouracil, inflammation

## Abstract

Intestinal mucositis is a common complication associated with 5-fluorouracil (5-FU), a chemotherapeutic agent used for cancer treatment. Cashew gum (CG) has been reported as a potent anti-inflammatory agent. In the present study, we aimed to evaluate the effect of CG extracted from the exudate of *Anacardium occidentale* L. on experimental intestinal mucositis induced by 5-FU. Swiss mice were randomly divided into seven groups: Saline, 5-FU, CG 30, CG 60, CG 90, Celecoxib (CLX), and CLX + CG 90 groups. The weight of mice was measured daily. After treatment, the animals were euthanized and segments of the small intestine were collected to evaluate histopathological alterations (morphometric analysis), levels of malondialdehyde (MDA), myeloperoxidase (MPO), and glutathione (GSH), and immunohistochemical analysis of interleukin 1 beta (IL-1β) and cyclooxygenase-2 (COX-2). 5-FU induced intense weight loss and reduction in villus height compared to the saline group. CG 90 prevented 5-FU-induced histopathological changes and decreased oxidative stress through decrease of MDA levels and increase of GSH concentration. CG attenuated inflammatory process by decreasing MPO activity, intestinal mastocytosis, and COX-2 expression. Our findings suggest that CG at a concentration of 90 mg/kg reverses the effects of 5-FU-induced intestinal mucositis.

## 1. Introduction

Cancer, a complex disease characterized by uncontrolled cell growth, is one of the main causes of morbidity and mortality in both developed and developing countries [1,2,3]. Chemotherapeutic agents can controlled the dissemination of several tumors and improve the quality of life in most patients with cancer. Currently, 5-fluorouracil (5-FU) is one of the major chemotherapeutic agents used for the treatment of cancer. 5-FU is a fluorinated pyrimidine with antimetabolite activity. However, it can cause side effects such as nausea, vomiting, diarrhea, myelosuppression, and intestinal mucositis [4,5,6,7,8,9].

Mucositis is initiated by basal cell lesions in the gastrointestinal tract, resulting in mucosal damage, intense inflammatory reaction, and consequent ulceration. It affects around 40% of patients treated with chemotherapeutic agents [10,11]. Intestinal mucositis can increase the risk of bacterial translocation and sepsis, thus impairing the continuity of anticancer treatment. 

Owing to the lack of efficacious therapeutic tools for the treatment of intestinal mucositis, new alternative therapeutics that can reduce the side effects of 5-FU, without impairing cancer treatment, have been investigated. 

Cashew gum (CG), a high-molecular weight complex heteropolysaccharide, is obtained from the exudate of cashew tree (*Anacardium occidentale* L.) through condensation of a large number of aldose and ketose molecules [12,13]. Its anti-inflammatory, antiulcerogenic, and antidiarrheal activity have been reported in previous studies [14,15,16,17]. In this study, we investigated the effect of CG on 5-FU-induced experimental intestinal mucositis. In addition, we evaluated its effect on inflammatory process and oxidative stress, as well as the involvement of cyclooxygenase 2 (COX-2).

## 2. Results

### 2.1. Weight Analysis

As expected, from the second day, all mice subjected to 5-FU-induced intestinal mucositis presented progressive weight loss, which was significant compared to the saline group (*p* < 0.05). Notably, only CG 60 pretreatment prevented weight loss induced by 5-FU (*p* < 0.05). CG 30 and CG 90 were unable to reverse 5-FU-induced weight loss (Figure 1).

### 2.2. Histopathological and Morphometric Analysis

The 5-FU group showed intense inflammatory cell infiltration, disruption of intestinal mucosal architecture, and a significant reduction in villus height, crypt depth, and villus/crypt ratio compared to saline group (*p* < 0.05) (Figure 2B). Notably, all doses of CG attenuated the effects induced by 5-FU (*p* < 0.05) (Figure 2C–H). Moreover, a significant increase in histopathological scores was found in 5-FU group (*p* < 0.05) compared to the saline group (Table 1). CG 90 pretreatment decreased the histopathological scores compared to the 5-FU group (*p* < 0.05). However, CG 30 and CG 60 pretreatment did not reduce histopathological scores.

### 2.3. Leukocyte Count

Analysis of leukocyte count in blood showed a significant decrease (*p* < 0.05) in the number of total leukocytes in 5-FU group compared to that in the saline group. In contrast, CG pretreatment reduced 5-FU-induced leukopenia (*p* < 0.05) (Figure 3A).

### 2.4. Myeloperoxidase Assay (MPO)

To investigate the effects of CG pretreatment on neutrophil recruitment in 5-FU-induced intestinal mucositis, we determined the activity of myeloperoxidase (MPO), a neutrophil marker. The 5-FU group presented a significant increase in MPO levels in the duodenum compared to the saline group (*p* < 0.05). CG (90 mg/kg) pretreatment decreased MPO levels in the duodenum of mice subjected to 5-FU-induced intestinal mucositis (*p* < 0.05), which in turn decreased polymorphonuclear leukocyte infiltration (Figure 3B).

### 2.5. Malondialdehyde (MDA) and Glutathione (GSH) Levels

To investigate the effect of CG pretreatment on 5-FU-induced oxidative stress in the duodenum, MDA and GSH levels (end products of oxidative stress) were evaluated. We found that 5-FU elevated MDA levels in the duodenum compared to the saline group (Figure 3C). CG (90 mg/kg) pretreatment reduced MDA levels compared to the 5-FU group (*p* < 0.05), thereby reducing 5-FU-induced oxidative stress (Figure 3C). Animals treated with 5-FU showed a significant (*p* < 0.05) decrease in GSH levels compared to the saline group. In contrast, CG (90 mg/kg) increased GSH levels compared to the 5- FU group (*p* < 0.05) (Figure 3D).

### 2.6. Mast Cell Concentration Analysis

To evaluate the effect of CG pretreatment on 5-FU-induced mastocytosis, the number of mast cells in the duodenum was measured. 5-FU (Figure 4B) increased the number of mast cells per field in the duodenum compared to the saline group (Figure 4A) (*p* < 0.05). CG (90 mg/kg) pretreatment (Figure 4C) decreased the number of mast cells compared to the 5-FU group (*p* < 0.05) (Figure 4D).

### 2.7. Effect of CG on Cyclooxygenase-2 Pathway in Histopathological and Morphometric Analyses

To investigate whether the effects of CG on reduction of 5-FU-induced intestinal injury, oxidative stress, and inflammation are mediated by cyclooxygenase-2 (COX-2) pathway, we blocked COX-2 by injecting celecoxib (CLX) in the presence or absence of CG in mice subjected to 5-FU-induced intestinal mucositis. Pretreatment with COX-2 blocker (CLX) (Figure 5D), as well as pretreatment with the combination of CLX and CG (90 mg/kg) (Figure 5E) prevented 5-FU-induced shortening of villus, cellular vacuolization, infiltration of inflammatory cells, edema, and loss of cellular architecture (Figure 5B).

CLX pretreatment decreased 5-FU-induced villus shortening (*p* < 0.05). In addition, pretreatment with the combination of CLX and CG (90 mg/kg) reverted villus shortening induced by 5-FU (*p* < 0.05). Moreover, the combination of CLX and CG (90 mg/kg) (Figure 5F) showed a greater effect on the recovery of duodenal villus in mice subjected to 5-FU-induced intestinal mucositis than pretreatment with CG 90 (Figure 5C) or CLX alone (Figure 5C) (*p* < 0.05).

### 2.8. Immunohistochemistry for the Detection of COX-2 and IL-1β

We investigated the effects of CG (90 mg/kg) in the presence or absence of CLX on COX-2 and IL-1β expression during 5-FU-induced intestinal mucositis through immunohistochemical analysis. 5-FU promoted intense immunostaining of COX-2 (Figure 6C) and IL-1β (Figure 6D) in the duodenal mucosa compared to the saline group (Figure 6A,B,K,L, *p* < 0.05). As shown in Figure 6E,F CG (90 mg/kg) pretreatment decreased immunostaining for COX-2 and IL-1β, respectively, compared to 5-FU group (Figure 6K,L, *p* < 0.05). Similarly, CLX alone (Figure 6G,H) or the combination of CLX and CG (90 mg/kg) (Figure 6I,J) decreased cell immunostaining forproinflammatory molecules in mice subjected to 5-FU-induced intestinal mucositis compared to the 5-FU group (Figure 6K,L, *p* < 0.05). Intense immunostaining was evidenced for the 5-FU lesion group, on the lamina followed by mild mucosal immunostaining, also was observed an absence of labeling in the epithelial cells.

## 3. Discussion

In the present study, we evaluated the effects of CG on intestinal mucositis induced by 5-FU and found that CG reversed the effects of 5-FU-induced intestinal mucositis at a concentration of 90 mg/kg. With the continuous expanded use of medicinal plants for the prevention and treatment of different pathologies worldwide, there has been increasing interest in the discovery of natural products with pharmacological effects. In Brazil, many herbal extracts are used in folk medicine to treat various digestive disorders, and several studies have documented the benefits of Brazilian plants in the prevention of gastrointestinal lesions [18,19].

A previous study reported that the extract of *Spondias pinnata*, belonging to the family Anacardiaceae, decreased histological severity scores and intestinal inflammation, and altered the mucosal architecture after chemotherapy. This indicated that *S. pinnata* extract could be an important pharmacological agent in promoting healing of damaged intestine after chemotherapy [20]. *A. occidentale* L. has been reported to possess diverse pharmacological properties. Its bark, leaves, and bark oil are used in anti-inflammatory and astringent preparations for the treatment of diarrhea [21]. Previous studies using the extracts of cashew tree bark have reported hypoglycemic [22,23], antioxidant and anti-inflammatory [24], antimicrobial [25], antihypertensive [26], and anticancer effects. It also showed beneficial effects in the treatment of gastritis, diarrhea, and wounds [22,27]. The anti-inflammatory activities and wound healing potential of cashew nuts have also been reported [28].

Weight loss is considered one of the common side effects of 5-FU chemotherapy. Therefore, body mass measure is one of the daily evaluated parameters to confirm the model of intestinal mucositis induced by 5-FU. Similar to our results, previous studies showed a decrease in body weight of animals after 5-FU-induced intestinal mucositis [29,30,31]. In this study, we showed that CG 60mg/kg decreased 5-FU-induced weight loss in mice, instead of the other CG doses which the loss weight was irreversible. Studies with probiotics and olmesartan showed that doses of these drugs were considered effective in reversing the harmful effects promoted by chemotherapy in mucositis model, after evaluation of histopathological parameters and inflammatory markers, however they were not effective in reverse weight loss as presented in ours studies [32,33].

Previous studies reported that 5-FU promoted decrease and vacuolization of intestinal villi, cryptic necrosis, infiltration of inflammatory cells, loss of cell architecture, and a decrease in villus/crypt ratio [34,35,36,37]. These findings are consistent with those of the present study. In addition, we found that CG at a concentration of 90 mg/kg was able to reduce the harmful effects of 5-FU on the duodenal mucosa. The anti-inflammatory effect of CG has been reported in skin lesions [15]. Araújo et al. [17] reported that 90 and 60 mg/kg of CG were effective in the treatment of acute inflammatory diarrhea. Similarly, 90 mg/kg was the best treatment concentration to prevent histopathological changes in 5-FU-induced intestinal mucositis.

Mucositis and myelosuppression are the main adverse effects related to 5-FU treatment [7,9,38]. In the present study, we showed that CG (90 mg/kg) attenuated leukopenia, the decrease in a number of total leukocytes, induced by 5-FU. Soares et al. [39] and Quaresma [40] demonstrated leukopenia in mice following a single administration of 5-FU (450 mg/kg).

The ulcerative phase of mucositis is characterized by loss of epithelial integrity of the mucosa, which facilitates a propitious environment for the invasion of bacteria. In addition, leucopenia may potentialize this process, resulting in bacteremia or sepsis [11,41,42]. We suggested that CG can be fundamental in preventing potential generalized bacterial infections during 5-FU treatment because it attenuated 5-FU-induced leukopenia. However, further investigations are needed to confirm this.

CG 90 prevented a 5-FU-induced increase in MPO levels in the duodenum. Similar to the present study, Bastos et al. [43], Justino et al. [44], Ávila et al. [45], Al-Asmari et al. [46], and Carvalho et al. [30] showed an increase in MPO activity after induction of intestinal mucositis by 5-FU. MPO has been used as a quantitative marker of neutrophil infiltration into various organs, including the gastrointestinal tract.

We showed that CG exerted antioxidant effects against intestinal mucositis by increasing GSH levels and decreasing MDA levels in the duodenum of mice subjected to 5-FU-induced intestinal mucositis. This finding is in agreement with the result of a previous study demonstrating that CG formulations exerted an antioxidant effect by decreasing MDA levels [47].

Our findings showed that CG reversed 5-FU-induced mastocytosis (increase in a number of mast cells) in the duodenum, indicating potent anti-inflammatory effects of CG in mice with 5-FU-induced intestinal mucositis. Previous studies have shown intense mastocytosis during intestinal mucositis induced by chemotherapeutics in mice [46,48]. The role of mast cells in the gastrointestinal tract is paradoxical, as their function depends on the mediators released and receptors activated. Mast cells contribute to intestinal homeostasis through immune protection, regulation of architecture and permeability of the epithelial barrier, and mucosal tissue remodeling through stimulation of fibroblast growth [49]. However, they are considered critical in the pathogenesis of inflammatory processes such as mucositis, because their overexpression consequently leads to amplification of inflammatory response caused by the selective release of mediators [50,51]. Besides exacerbation of inflammation, mastocytosis in the gastrointestinal tract has been reported to culminate in alteration of architecture and impairment of the gastrointestinal barrier, such as villus enlargement and changes in crypt size in the small intestine [52,53].

In the present study, we evaluated the effect of CG on COX-2 pathway by pretreating mice subjected to intestinal mucositis with a combination of CLX (a COX-2 blocker) and CG (90 mg/kg). In addition, we investigated whether the protective effect of CG on morphometric and histopathological changes induced by 5-FU was related to COX-2 inhibition. We found that the combination of CLX and CG (90 mg/kg) completely reverted 5-FU-induced decrease in villi, cryptic necrosis, inflammatory cell infiltration, and loss of cellular architecture in the duodenum. Our findings demonstrated that pretreatment with the combination of CG and CLX during 5-FU-induced intestinal mucositis was more effective in reversing histopathological effects than treatment with CLX alone, a commercially available nonsteroidal anti-inflammatory drug that acts as a selective inhibitor of COX-2. Short et al. [54] suggested that low doses of CLX can be used therapeutically for the protection of the intestinal barrier in patients with inflammatory bowel disorders, because of its ability to reduce COX-2 expression. Javle et al. [55] found that the combination of irinotecan (CPT-11) and CLX resulted in antitumor effects, with improvement in irinotecan-induced diarrhea and lethality. Furthermore, CG (90 mg/kg) alone or in combination with CLX was able to decrease 5-FU-induced COX-2 and IL-1β immunostaining in the duodenum. It is known that IL-1β is produced by macrophages, monocytes, and glial cells. This proinflammatory cytokine induces the expression of inflammatory mediators, such as COX-2 with subsequent release of prostaglandins, and the onset of primary tissue damage and progression [56,57,58]. The gene expression and tissue levels of IL-1β are correlated with intestinal mucosal injury induced by chemotherapy [59].

The study and use of natural polysaccharides in inflammatory bowel diseases, especially mucositis, is a current reality and future promise for the development of effective drugs in the treatment of 5-FU-induced intestinal mucositis. The present study has however a limitation. As intestinal inflammation is a complex process involving multiple mechanisms of activation and maintenance of the inflammatory process, the hypothetical model of action of CG on 5-FU-induced intestinal mucositis in mice proposed using the results of the present study (Figure 7) may be inadequate. Further studies are needed to elucidate the other possible mechanisms of CG action in intestinal mucositis.

## 4. Materials and Methods

### 4.1. Animals

The animals were obtained from the Department of Surgery of the Federal University of Ceara (UFC) were used. The male Swiss mice (25–30 g) were housed in polypropylene cages, lined with wood, in a controlled environment with a temperature of 23 ± 2 °C, in a cycle of 12 h light/12 h dark, with free access to water and standard feed. The procedures and experimental protocols were approved by the Ethics Committee on Animal Use (n° 208/16) from the Federal University of Piaui (CEUA/UFPI).

### 4.2. Drugs and Plant Materials

Two drug drugs were used for mucositis induction and treatment, respectively: 5-FU (FauldFluor^®^, Libbs, Sao Paulo, Brazil) celecoxib (CLX- Celebra^®^, Pfizer, Sao Paulo, Brazil). Raw samples of CG were collected in 2013 by the Biotechnology and Biodiversity Center Research–BIOTEC, Parnaíba, Brazil, from the trunk of native cashew trees (*A. occidentale* L.) in Ilha Grande de Santa Isabel, Piauí, Brazil (Latitude, decimal degrees S-2.8242; Longitude, decimal degrees W-41.7331). The tree was identified and a voucher specimen, voucher number 52, was deposited at the HDELTA herbarium (Federal University of Piauí, Parnaíba, Piauí, Brazil).

### 4.3. Extraction and Purification of Cashew Gum

The CG was purified with sodium salt, as previously described [60]. Bark free nodules were selected and dissolved at a final concentration of 5% (*w*/*v*) in distilled water. The pH of the solution was adjusted to approximately 7.0. The clear solution was successively filtered through sintered glass and the heteropolysaccharide was precipitated with ethanol [61,62].

### 4.4. Induction of Experimental Intestinal Mucositis

The experimental intestinal mucositis model in Swiss Mice was induced as described by Soares et al. [39]. The 5-FU (450 mg/kg) was given intraperitoneally (i.p) in a single dose on the first day of the experimental protocol. Three days of treatment with CG were performed. The mice were pretreated daily with oral CG (30, 60, 90 mg/kg), 1h before the injection of 5- FU, after then the same oral doses were performed once a day. After four days of treatment, the animals were euthanized by ketamine (270 mg/kg) and xylazine (15 mg/kg) and the intestinal samples were collected. The body weight of mice was measured daily before the treatment administered to confirm the experimental model of intestinal mucositis induced by 5-FU. In this study, the mice was randomly allocated in seven groups (*n* = 6): Saline (NaCl 0.9%), 5-FU (5-FU + NaCl 0.9%), CG 30 (5-FU + CG 30 mg/kg), CG 60 (5-FU + CG 60 mg/kg), CG 90 (5-FU + CG 90 mg/kg), CLX (5-FU + celecoxib 7.5 mg/kg, i.p), CLX + CG 90 (5-FU + celecoxib 7.5 mg/kg, i.p + CG 90 mg/kg). To investigate the participation of COX-2 on the effects of treatment with CG during intestinal mucositis induced by 5-FU, COX-2 was blocked by celecoxib in an independent experiment. The mice were treated with celecoxib and CG 90 in combination or alone for three days.

### 4.5. Histopathological and Morphometric Analysis

After euthanasia, duodenum samples were obtained and fixed in 10% formaldehyde for performing the histopathological and morphometric analysis [63,64]. These samples were embedded in paraffin, sectioned at 4 µm and stained with hematoxylin and eosin (H&E). A blinded and randomized histopathological analysis was performed by an experienced histopathologist to assess the severity of mucositis using a scores system [65], the tissues were ranging from 0 (absence of lesion/normal histological findings) to 3 (maximum lesion degree), indicating shortened villi with vacuolized cells, necrosis of crypts, intense infiltration of inflammatory cells, vacuolization and edema in the mucosal layer and muscular layer with edema, vacuolization and neutrophil infiltrate. The effective concentration of CG in the treatment of mucositis was determined following the histological analysis.

### 4.6. Leukocyte Count

Mice were anesthetized with a combination of anesthetics (xylazine 10m/kg and ketamine 80 mg/kg) and, a peripheral blood sample was collected from the ocular artery, and was diluted in the liquid of Turk at a ratio of 20 μL of blood to 380 μL of solution. The total leukocytes were counted using a Neubauer chamber [66], and the results were expressed as a total number of leukocytes per mm3 of blood.

### 4.7. Dosage of Malondialdehyde (MDA)

The MDA is a product of lipid peroxidation frequently used as a marker of oxidative stress. Briefly, the intestinal samples were homogenized (10%) with potassium phosphate buffer (1.15%). Then, were added phosphoric acid (1%) and thiobarbituric acid (0.6%) to the homogenates and incubated at (100 °C, for 45 min) followed by the addition of 1.5 mL *n*-butanol. The supernatant was obtained and measured after centrifugation (5000 rpm, for 10 min). The results were expressed as nMol of MDA/mg of tissue by 535 nm absorbance measure [67,68].

### 4.8. Concentration of Glutathione (GSH) 

The concentration of GSH in the duodenal samples was performed according to the method described by Sedlak and Lindsay [69]. The levels of nonprotein sulfhydryl groups (NPSH) were determined from 50 to 100 mg of the intestinal mucosa of each animal. The tissues were homogenized in 1 mL of 0.02M EDTA for each sample. Aliquots of 100 μL of the homogenate were mixed with 80 μL of distilled water and 20 μL of 50% trichloroacetic acid (TCA) for precipitation of proteins. The tubes were centrifuged for 15 min at 3000 rpm at 4 °C. A total of 100 μL of the supernatant was added to 200 μL of 0.4 M Tris buffer (pH 8.9) and 5 μL of 0.01 M 5,5-dithiobis-(2-nitrobenzoic acid) (DTNB, Sigma Aldrich, St. Louis, Missouri, USA). The mixture was then homogenate for 3 min and the absorbance was read at 412 nm. Results were expressed as micrograms of NPSH groups per milliliter of homogenate (μg/mL).

### 4.9. Myeloperoxidase Assay (MPO)

MPO activity was determined by the technique described by Bradley et al [70]. Briefly, the duodenum segments (50–100 mg) were homogenized in 1 mL of potassium buffer containing 0.5% hexadecyltrimethylammonium bromide (HTAB), then centrifuged (4000 rpm, 7 min, 4 °C). The MPO activity was analyzed by measuring the absorbance at 450 nm using diisocyanate dihydrochloride and 1% hydrogen peroxide in the resuspended pellet. The results were recorded as MPO units per mg tissue.

### 4.10. Mast Cell Analysis

The paraffin blocks with duodenum samples were processed for toluidine blue staining to identified mast cells, according to Michalany [71]. The slides were deparaffinized with xylene, incubated for 3 min with toluidine blue solution (1 g of toluidine blue dissolved in 70% ethanol), washed three times in distilled water, dehydrated, and mounted. The total number of mast cells were counted manually, considering four specimens per group and ten fields per slide. The results were expressed as the mean of 10 fields in each group.

### 4.11. Immunohistochemistry for the Detection of COX-2 and IL-1β

Duodenal sections were deparaffinized with oven insertion (60 °C) and three cycles of xylol immersion for 5 min each. Then, the sections were rehydrated in decreasing alcohol concentrations (100, 90, 80 and 70%). The histological sections were then washed with distilled water for 10 min and the antigenic recovery in citrate buffer (pH 7.0, DAKO^®^, Sao Paulo, Brazil) was carried out for 20 min in the water bath (95 °C). The slides were then washed with phosphate-buffered saline solution (PBS) for 5 min at room temperature. Following, endogenous peroxidase blockade with 3% hydrogen peroxide solution (H_2_O_2_) was performed for 30 min. The sections were then incubated overnight with goat anti-COX-2 primary antibody (SantaCruz^®^, Dallas, TX, USA), and rabbit anti-IL-1β (SantaCruz^®^, Dallas, TX, USA) diluted in antibody diluent (1:100) for 60 min, respectively. After the slides were washed with PBS and incubated with rabbit IgG (GBI Labs^®^, Bothell, WA, USA) secondary antibody diluted (1:400) for 30 min. For revelation, the sections were incubated with the streptavidin conjugated peroxidase complex (ABC complex) for 30 min and chromogen 3,3′diaminobenzidine peroxide, DAB (DAKO^®^, Sao Paulo, Brazil), followed by counterstaining with hematoxylin (DAKO^®^, Sao Paulo, Brazil), for 10 min. Negative controls were processed simultaneously as described above, with the primary antibody being replaced for antibody diluent. The procedures were performed in an automated manner using Autostainer Plus (DAKO^®^, Sao Paulo, Brazil). To assess COX-2 immunostaining, quantification was performed by immunolabelled cells with the aid of Image J software. For IL-1β immunostaining images, quantification was performed by measuring the % immunolabelled area with the aid of Adobe Photoshop 10. All images were captured with the aid of an optical microscope to the image acquisition system (LEICA, Wetzlar, HE, Germany ).

### 4.12. Statistical Analysis

Quantitative results were expressed as mean ± standard error of the mean (SEM) and the qualitative data (histological scores) were pointed scores and expressed by the median ± minimum and maximum. The results with a parametric distribution were analyzed by Analysis of Variance (ANOVA) followed by post hoc test Tuckey through the program GraphPad Prism version 6.0 (GraphPad Software Inc., La Jolla, CA, USA). The data obtained from non-parametric distribution were analyzed using Kruskal-Wallis test followed by Dunn’s (multiple comparisons). Values of *p*-value < 0.05 were considered statistically significant.

## 5. Conclusions

In summary, CG decreased inflammation, oxidative stress, and intestinal injury induced by 5-FU in the duodenum. The effects of CG were found to be related to COX-2 pathway. The concomitant administration of CG and CLX completely reverted COX-2 and IL-1β immunostaining markers and intestinal injury induced by 5-FU. Thus, we suggest that CG has potential application in the development of novel drugs against intestinal mucositis due to antineoplastic agents. Additionally, we recommend further studies to elucidate the molecular mechanisms related to the effects of GC under pro-inflammatory cytokines expression as well as other possible mechanisms of action involved in the protective effect of CG on chemotherapy-induced intestinal mucositis.

## Figures and Tables

**Figure 1 pharmaceuticals-12-00051-f001:**
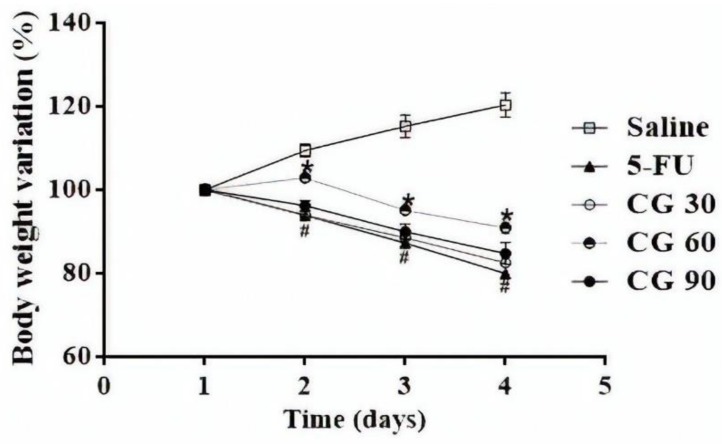
Body weight variation in mice subjected to intestinal mucositis (5-FU, 450 mg/kg, ip, single dose) and treated with CG (30, 60 and 90 mg/kg for 4 days). The results are expressed as the mean ± SEM of the weight evaluation percentage of the initial weight, of a minimum of 6 animals per group. Two-way ANOVA followed by the Tukey’s test were used for the statistical analysis, where # *p* < 0.05 vs. saline and * *p* < 0.05 vs. 5-FU.

**Figure 2 pharmaceuticals-12-00051-f002:**
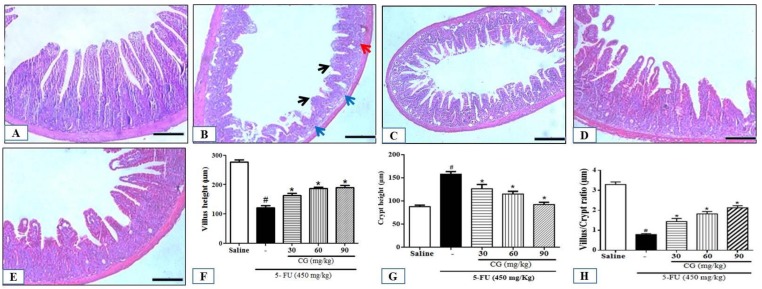
Histopathological analysis. (**A**) Saline; (**B**) 5-FU; (**C**) CG 30; (**D**) CG 60; (**E**) CG 90. 5 FU induced inflammatory cell infiltrate (red arrow), decreased intestinal villi (black arrow), loss of intestinal crypt architecture (blue arrow). Pretreatment with CG (30, 60 and 90 mg/kg) decreased the inflammatory infiltrate and prevented the shortening of the villi (**F**), increased crypt depth (**G**) and decreased villus/crypt ratio (**H**), with greater reversion of the 5-FU effect in the CG 90 + 5-FU group. All panels were obtained on the 100 μm scale (×200). Values were expressed as mean ± SEM. One-way ANOVA followed by the Tukey’s test were used for the statistical analysis was used, where # *p* < 0.05 vs. saline group and * *p* < 0.05 vs. group 5-FU.

**Figure 3 pharmaceuticals-12-00051-f003:**
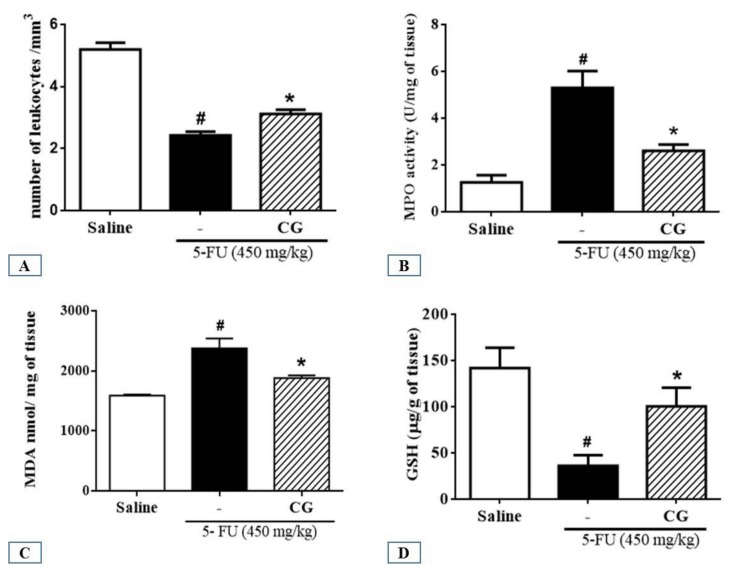
(**A**) Total leukocyte count; (**B**) Activity of myeloperoxidase (MPO); (**C**) Level of malondialdehyde (MDA); (**D**) concentration of glutathione (GSH). Values were presented as mean ± SEM. For the statistical analysis, one-way ANOVA followed by Tukey’s test was used, where # *p* < 0.05 vs. saline group and * *p* < 0.05 vs. group 5-FU.

**Figure 4 pharmaceuticals-12-00051-f004:**
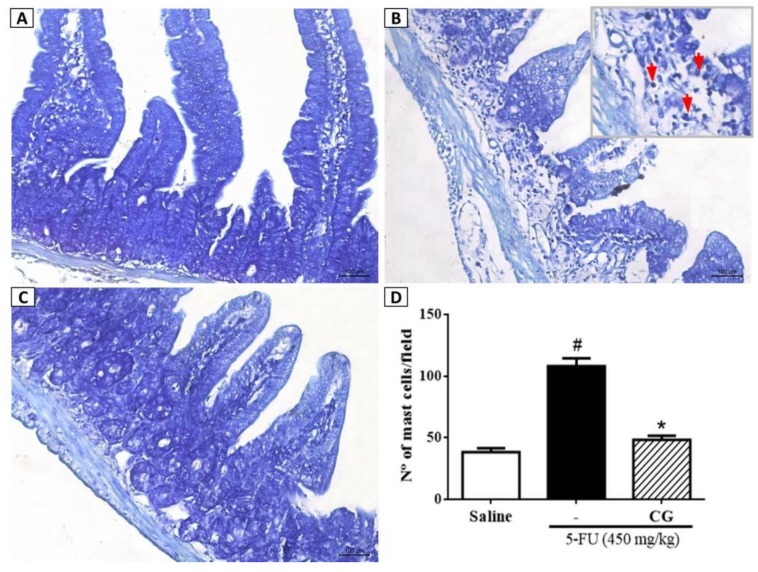
Mast cell counts in the duodenum samples. In (**B**) demonstrates that 5-FU promoted increased mast cell counts (red arrows) when compared to saline group (**A**). CG 90 (**C**) reversed the 5-FU-induced mastocytosis. All the panels were obtained at ×400 magnification. (**D**) Values were presented as mean ± SEM of the number of mast cells per field. For the statistical analysis, tone-way ANOVA followed by Tukey’s test was used, where # *p* <0.05 vs. saline group and * *p* < 0.05 vs. group 5-FU.

**Figure 5 pharmaceuticals-12-00051-f005:**
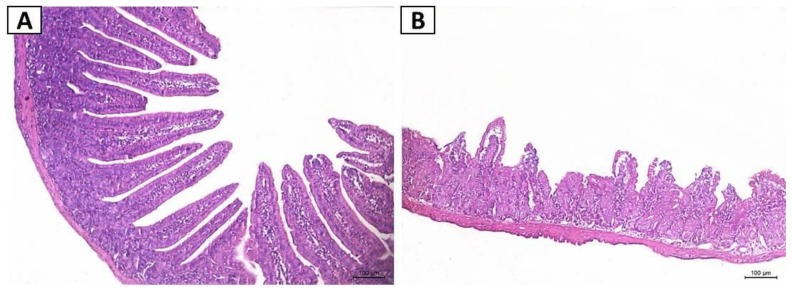
Effect of GC on the cyclooxygenase-2 pathway (COX-2) in histopathological and morphometric. The histopathological analysis represented by the groups (**A**) Saline; (**B**) 5-FU; (**C**) CG 90; (**D**) Celecoxib (CLX); (**E**) CLX and CG 90, as well as morphometric analysis of villus height (**F**) showed that 5-FU caused a decrease in villi and loss of cellular architecture when compared to the saline group. C, D, and E reversed the effect of 5-FU. All panels were obtained at ×200 magnification. Values were expressed as mean ± SEM for villi height in μm. For statistical analysis, one-way ANOVA followed by Tukey’s test was used, where # *p* < 0.05 vs. saline group, * *p* < 0.05 vs. group 5-FU, ** *p* < 0.05 vs. group CLX.

**Figure 6 pharmaceuticals-12-00051-f006:**
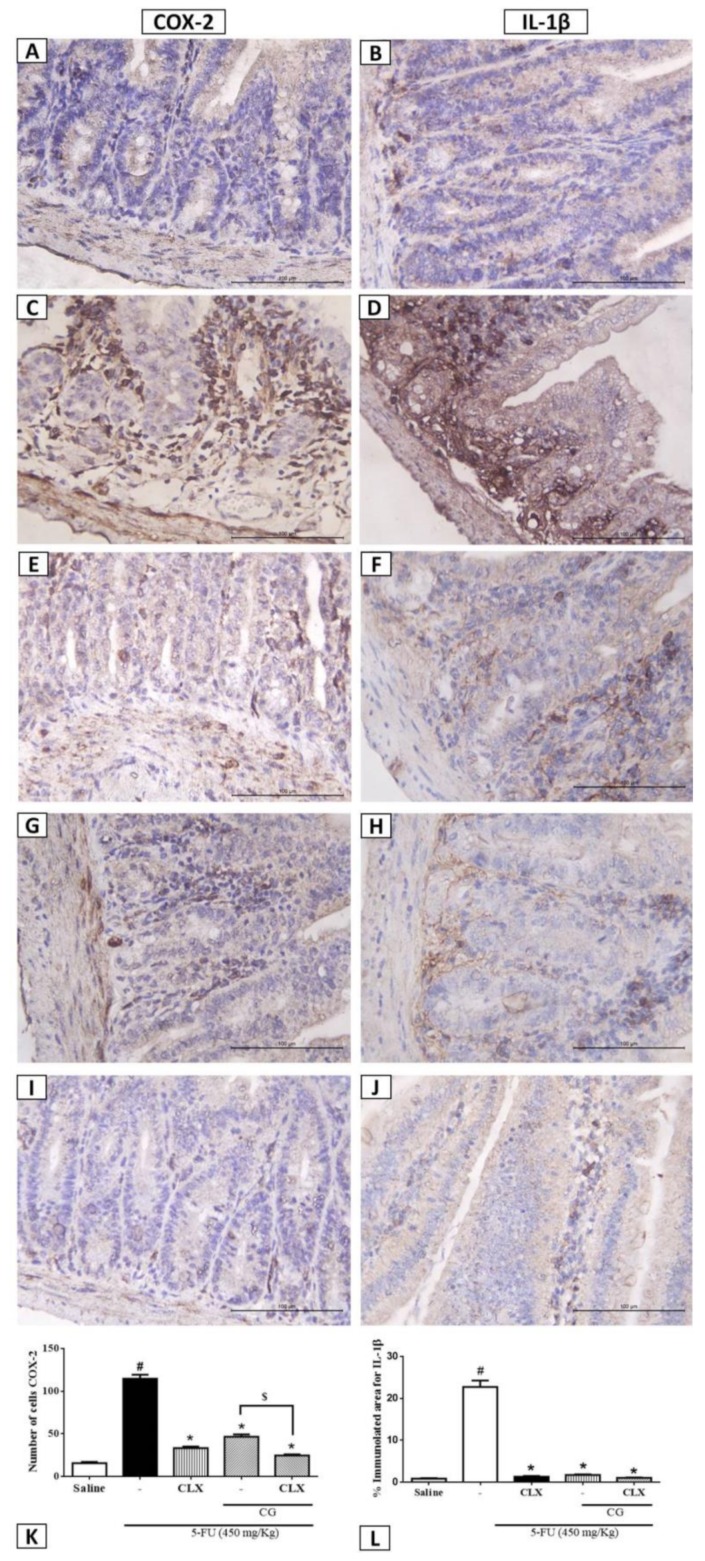
Immunohistochemistry analysis for COX-2 and IL-1β. (**A**,**B**) Saline; (**C**,**D**) 5-FU; (**E**,**F**) CLX; (**G**,**H**) CG 90; (**I**,**J**) CLX + CG 90. (**K**) a number of cells immunolabelled for cox-2. (**L**) % immunolabelled for IL-1β. Values were expressed as mean ± SEM. For statistical analysis, one-way ANOVA followed by Tukey´s test was used, where # *p* < 0.05 vs. saline group, * *p* < 0.05 vs. group 5-FU, $ *p* < 0.05 vs. group CLX.

**Figure 7 pharmaceuticals-12-00051-f007:**
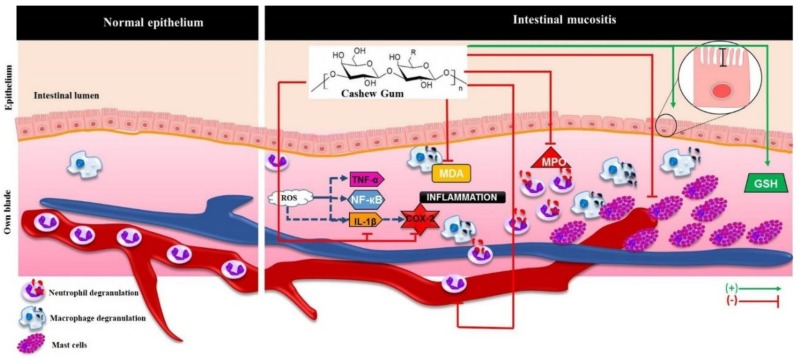
Hypothetical model of action of CG in intestinal mucositis induced by 5-FU. CG prevented the 5-FU-induced injury intestinal through inhibition of MDA formation, neutrophil recruitment (decreasing MPO levels), mast cells activation and IL-1β and COX-2 immunostaining marker and inhibited leucopenia. CG also stimulate villus enlargement and increase levels of GSH, an antioxidant. ROS: Reactive Oxygen Species; TNF-α: Tumor Necrosis Factor-alpha; NF-κB: transcription factor nuclear kappa b; IL-1β: Interleukin 1 beta; COX-2: Cyclooxygenase 2; MDA: Malondialdehyde; MPO: Myeloperoxidase; GSH: Reduced glutathione. Arrows green (stimulate / increase), red arrows (inhibit).

**Table 1 pharmaceuticals-12-00051-t001:** Histopathological scores of mice subjected to 5-FU-induced intestinal mucositis and pretreated with CG.

Groups	Scores
Saline	0 (0-0)
5-FU	2 (1-3) ^#^
CG 30	3 (3-3)
CG 60	1 (1-2)
CG 90	1 (0-1) *

Values were expressed as median, where # *p* < 0.05 vs. saline and * *p* < 0.05 vs. 5-FU (*n* = 6/group). The data was analyzed by the Kruskal-Wallis test followed by the Dunns multiple comparisons test.

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
