# Peer review of "Protective Effect of Cashew Gum (Anacardium occidentale L.) on 5-Fluorouracil-Induced Intestinal Mucositis"

_pharmaceuticals, 2019, doi:10.3390/ph12020051_

Round 1

Reviewer 1 Report

The purpose of this study was the effect of Cashew gum to prevent intestinal mucositis induced by 5-FU. They analyzed inflammatory process and oxidative stress, as well as the involvement of COX-2. They used different concentration of GC and they demonstrated that CG 90 prevented 5-FU-induced intestinal mucositis decreasing oxidative stress and inflammatory process

They conclude that CG at a concentration of 90 mg/kg reverses the effects of 5-FU-induced intestinal mucositis.

The study design is adequate to address the scientific questions raised in the paper. However, the authors should address some points:

-They conclude that CG 90 prevented intestinal mucositis however and that CG decreased 5-FU-induced weight loss in mice. However, in the figure 1, mice treated with CG 90 presented progressive weight loss. The only significant effect is with CG 60. However, CG 60 doesn´t have effect to prevent mucositis. these results are contradictory. The authors should explain the reason of this.

-The authors should include the GSH/GSSG ratio considering that the GSH/GSSG ratio is a good indicator of oxidative stress and no GSH alone.

-Page 6, the authors indicate they measure IL-1 beta expression by immunohistochemistry. However, by this technique they only can measure levels of Il-1 and no expression.

-The authors conclude that CG decreased IL-1 beta expression. However, to achieve this conclusion they should me measure IL-1 beta by PCR or western blot.

Author Response

Response to Reviewer 1 Comments

 Point 1: They conclude that CG 90 prevented intestinal mucositis however and that CG decreased 5-FU-induced weight loss in mice. However, in the figure 1, mice treated with CG 90 presented progressive weight loss. The only significant effect is with CG 60.

However, CG 60 doesn´t have effect to prevent mucositis. these results are contradictory. The authors should explain the reason of this. 

Response 1: The considerations are fully pertinent and were fully considered for our group.

We know the relevance of studies to minimize the side effects of chemotherapies in cancer treatment. The intestinal mucositis is defined by the presence of ulcerative and inflammatory lesions in the mucosa of the gastrointestinal tract. The mucositis induced by chemotherapy is evidenced in some experimental models by some set of parameters. Into clinical evaluations, the progressive weight loss is one of the most evident, as well as a hematological parameter as pronounced leukopenia and also histopathological evaluation that evidences the characteristic tissue damage. Into your data, we considered the set of data that together indicate the CG treatment can reduce the mucositis induced by 5-FU in the experimental model. Thus, clinical evaluation, though important, cannot be considered as a single and predominant parameter in the determination of the protective effect of GC in mucositis. For this, other parameters as oxidative stress and leukocyte infiltration were evaluated, as well as evaluating the reduction of villus height and crypts in the small intestine that is marked features of damaged tissues. Others studies performed by Gerhard et al. (2017) and Araújo et al. (2015) can support our conclusions about the characterization of a protective effect for mucositis model. In their study, Gerhard et al. (2017) and Araújo et al. (2015) also did not observe a full reversal of weight loss assessing the effect of chemotherapy on weight loss, after pretreatment with probiotic and Olmesartan, respectively. However, the drugs tested by these authors indicated protective effects considering other parameters evaluated, such as reduction of proinflammatory cytokines, immunosuppression and improve tissue architecture and reduction of characteristic histological damage.

Point 2: The authors should include the GSH/GSSG ratio considering that the GSH/GSSG ratio is a good indicator of oxidative stress and no GSH alone.

Response 2: Considering the biochemistry parameter evaluated, the antioxidant defense system includes enzymatic and non-enzymatic mechanisms. Among the endogenous enzymatic antioxidants are: superoxide dismutase (SOD), glutathione peroxidase (GSH-Px), glutathione reductase (GSH-Rd), catalase and superoxide reductase; and Thioredoxin, Melatonin and Reduced Glutathione (GSH) among non-enzymatic antioxidants. GSH is considered the most important non-enzymatic antioxidant, Glutathione and the intracellular redox state (GSH / GSSG levels) of the cell is often used as a marker of the antioxidant capacity of cells. In many inflammatory diseases in the gastrointestinal tract, depletion of intracellular GSH levels is present in concomitant with the induction of inflammatory mediators and proinflammatory cytokines (RAHMAN; MACNEEE, 2000; GAURAV et al., 2012; SCHMITT et al., 2015). Thus, the evaluation of intracellular GSH levels illustrates the panorama of antioxidant capacity against an inflammatory process. When high levels of GSH suggest a protective capacity for oxidative stress, inflammation, and apoptosis, the decrease in GSH levels suggests a low antioxidant effect and/or exacerbation of oxidative stress/inflammation (CHENG et al., 2017). Corroborating these findings, we have evaluated the levels of malondialdehyde (MDA), which are commonly the two parameters widely used in the scientific environment seen in studies by Al-Asmari et al. (2015); Dos Santos-Filho et al. (2016); Alvarenga et al. (2016), and which are also part of the scope of the journal in question as seen in the work of Carneiro et al. (2018).

Point 3: Page 6, the authors indicate they measure IL-1 beta expression by immunohistochemistry. However, by this technique they only can measure levels of Il-1 and no expression. The authors conclude that CG decreased IL-1 beta expression. However, to achieve this conclusion they should me measure IL-1 beta by PCR or western blot.

Response 3: The histological features to inflammation in the intestinal mucosa are characterized by an infiltration of neutrophils and macrophages, which when activated are important producers of cytokines and other inflammatory mediating agents. Among the proinflammatory cytokines, the IL-1 interleukins family (α and β) play key roles in the inflammatory response through the modulation of many others cytokines (ANDUS et al., 1997). It is notorious that the quantitative molecular techniques for gene and protein expression (such as PCR and Western Blot) can optimize the levels of IL-1β detection, allowing with this union to increase the sensitivity, specificity, and possibility of indicating the in situ localization in the tissue of the protein under study instead Immunohistochemistry (IHQ). However, the IHQ presents as a differential the possibility of the track the areas with protein expression (marks) in the tissue (SKOG et al., 2014). Some studies corroborate with this characteristic of IHQ studies. Yougmamn et al. (1993) evidenced the synthesis and expression of IL-1 in the intestine and absent in epithelial cells. Olson et al. (1993) and Maihofner et al. (2003) demonstrated a predominance of IL-1β in the submucosa and mucosa with a marked emphasis on the duodenal crypt region, reinforced by the intensity and quantity of IL-1β-labeled cells in the intestinal regions. Also, some studies using qualitative and quantitative technics to validate immunohistochemical assays show high levels of agreement for the results. In studies using IHQ and PCR, Sinn et al. (2017) evidenced agreement in their results as well as Prihantono et al. (2017) using the same techniques to evaluate the ki-67 expression. Cleary, quantitative assays as PCR are an important alternative to conventional visual estimation, widely used in research methods, especially to histologic studies.

Conclusion

As demonstrated in our study the findings indicate the protective effect of CG in the model studied. When immunolabelled IL-1β positive, the intestinal samples showed intense labeling in lamina propria and part of the mucosa cells, especially in the 5-FU lesion group, when compared to CG 90 treated group. Considering the findings and the absence of quantitative data for the cytokines expression we will depict in our article only an increase or decrease of immunolabelled cells (% of immunolabelled cells) and labeling intensity as well as predominant in situ localization of this immunoblot in a similar manner as already discussed by Eriksson et al. (1999) and Bertevello et al. (2005) when presenting results of immunohistochemistry for IL-1β and other chewers.

References

AL-ASMARI, Abdulrahman Khazim et al. Ascorbic acid attenuates antineoplastic drug 5-fluorouracil induced gastrointestinal toxicity in rats by modulating the expression of inflammatory mediators. Toxicology reports, v. 2, p. 908-916, 2015.

ALVARENGA, Elenice M. et al. Carvacrol reduces irinotecan-induced intestinal mucositis through inhibition of inflammation and oxidative damage via TRPA1 receptor activation. Chemico-biological interactions, v. 260, p. 129-140, 2016.

ANDUS, T. et al. Imbalance of the interleukin 1 system in colonic mucosa—association with intestinal inflammation and interleukin 1 receptor agonist genotype 2. Gut, v. 41, n. 5, p. 651-657, 1997.

BERTEVELLO, Pedro L. et al. Immunohistochemical assessment of mucosal cytokine profile in acetic acid experimental colitis. Clinics, v. 60, n. 4, p. 277-286, 2005.

BHATTACHARYYA, Asima et al. Oxidative stress: an essential factor in the pathogenesis of gastrointestinal mucosal diseases. Physiological reviews, v. 94, n. 2, p. 329-354, 2014.

CARNEIRO, José et al. Gastroprotective Effects of Sulphated Polysaccharides from the Alga Caulerpa mexicana reducing ethanol-induced gastric damage. Pharmaceuticals, v. 11, n. 1, p. 6, 2018.

CHENG, Shao-Bin et al. Changes of oxidative stress, glutathione, and its dependent antioxidant enzyme activities in patients with hepatocellular carcinoma before and after tumor resection. PloS one, v. 12, n. 1, p. e0170016, 2017.

DE ARAÚJO, Aurigena Antunes et al. In a methotrexate-induced model of intestinal mucositis, olmesartan reduced inflammation and induced enteropathy characterized by severe diarrhea, weight loss, and reduced sucrose activity. Biological and Pharmaceutical Bulletin, v. 38, n. 5, p. 746-752, 2015.

DOS SANTOS FILHO, Edvande Xavier et al. Curcuminoids from Curcuma longaL. reduced intestinal mucositis induced by 5-fluorouracil in mice: Bioadhesive, proliferative, anti-inflammatory and antioxidant effects. Toxicology Reports, v. 3, p. 55-62, 2016.

ERIKSSON, C. et al. Immunohistochemical localization of interleukin-1β, interleukin-1 receptor antagonist and interleukin-1β converting enzyme/caspase-1 in the rat brain after peripheral administration of kainic acid. Neuroscience, v. 93, n. 3, p. 915-930, 1999.

GAURAV, K. et al. Glutamine: A novel approach to chemotherapyinduced toxicity. Indian Journal of Medical and Paediatric Oncology, v. 33, n. 1, p. 13-20, 2012.

GERHARD, Dayana et al. Probiotic therapy reduces inflammation and improves intestinal morphology in rats with induced oral mucositis. Brazilian oral research, v. 31, 2017.

KART, Asim et al. The therapeutic role of glutathione in oxidative stress and oxidative DNA damage caused by hexavalent chromium. Biological trace element research, v. 174, n. 2, p. 387-391, 2016.

MAIHÖFNER, Christian et al. Expression of cyclooxygenase-2 parallels expression of interleukin-1beta, interleukin-6 and NF-kappaB in human colorectal cancer. Carcinogenesis, v. 24, n. 4, p. 665-671, 2003.

OLSON, Allan D.; AYASS, Mouhib; CHENSUE, Stephen. Tumor necrosis factor and IL-1 beta expression in pediatric patients with inflammatory bowel disease. Journal of pediatric gastroenterology and nutrition, v. 16, n. 3, p. 241-246, 1993.

PRIHANTONO, Prihantono et al. Ki-67 expression by immunohistochemistry and quantitative real-time polymerase chain reaction as predictor of clinical response to neoadjuvant chemotherapy in locally advanced breast cancer. Journal of oncology, v. 2017, 2017.

RAHMAN, I.; MACNEE, W. Oxidative stress and regulation of glutathione in lung inflammation. European Respiratory Journal, v. 16, n. 3, p. 534-554, 2000.

SCHMITT, Bernard et al. Effects of N-acetylcysteine, oral glutathione (GSH) and a novel sublingual form of GSH on oxidative stress markers: A comparative crossover study. Redox biology, v. 6, p. 198-205, 2015.

SINN, Hans-Peter et al. Comparison of immunohistochemistry with PCR for assessment of ER, PR, and Ki-67 and prediction of pathological complete response in breast cancer. BMC cancer, v. 17, n. 1, p. 124, 2017.

SKOG, Oskar; INGVAST, Sofie; KORSGREN, Olle. Evaluation of RT-PCR and immunohistochemistry as tools for detection of enterovirus in the human pancreas and islets of Langerhans. Journal of Clinical Virology, v. 61, n. 2, p. 242-247, 2014.

YOUNGMAN, Kenneth R. et al. Localization of intestinal interleukin 1 activity and protein and gene expression to lamina propria cells. Gastroenterology, v. 104, n. 3, p. 749-758, 1993.

Reviewer 2 Report

The manuscript "Protective effect of cashew gum (Anacardium occidentale L.) on 5-fluorouracil-induced intestinal mucositisshows new and interesting results, and ii appears as a good work well structured. The authors have well synthesized the literature in the introduction and the topic is original and of great interest. The article is well-organized and easy to understand, The conclusion that the cashew gum could be helpfull for the avversive effect of the chemiotherapic 5-fluorouracil is very interesting and promising in the field of cancer therapy. However, few typing errors in the text and references could be corrected for the final version.

Author Response

Point 1 Protective effect of cashew gum (Anacardium occidentale L.) on 5-fluorouracil-induced intestinal mucositis"  shows new and interesting results, and ii appears as a good work well structured. The authors have well synthesized the literature in the introduction and the topic is original and of great interest. The article is well-organized and easy to understand, The conclusion that the cashew gum could be helpfull for the avversive effect of the chemiotherapic 5-fluorouracil is very interesting and promising in the field of cancer therapy. However, few typing errors in the text and references could be corrected for the final version.

Response 1: The considerations are fully pertinent and were fully considered for our group.

Orthographic corrections were made throughout the text, however, for the English revision and correction process, the article was evaluated by a company specialized in the area for this purpose.

Round 2

Reviewer 1 Report

The authors have addressed all the recommendations.